# CAN GRADIENT CLIPPING MITIGATE LABEL NOISE?

**Aditya Krishna Menon, Ankit Singh Rawat, Sashank J. Reddi, Sanjiv Kumar**
Google Research
New York, NY USA
`{adityakmenon,ankitsrawat,sashank,sanjivk}@google.com`

## ABSTRACT

Gradient clipping is a widely-used technique in the training of deep networks, and is generally motivated from an *optimisation* lens: informally, clipping controls the dynamics of iterates, thus enhancing the rate of convergence to a local minimum. This intuition has been made precise in a line of recent works, which show that suitable clipping can yield significantly faster convergence than vanilla gradient descent. In this paper, we propose a new lens for studying gradient clipping, namely, *robustness*: informally, one expects clipping to mitigate the effects of noise, since one does not overly trust any single sample. Surprisingly, we prove that for the common problem of *label noise* in classification, standard gradient clipping does *not* in general provide robustness. On the other hand, we show that a simple variant of gradient clipping *is* robust, and is equivalent to suitably modifying the underlying loss function. As a special case, this yields a simple, noise-robust modification of the standard cross-entropy loss which performs well empirically.

## 1 INTRODUCTION: TWO FACES OF GRADIENT CLIPPING

Gradient clipping is a widely-used technique in the training of deep networks (Mikolov, 2012; Pascanu et al., 2012; 2013; Mnih et al., 2015; Gehring et al., 2017; Merity et al., 2018; Peters et al., 2018), and in its simplest form, involves capping the global parameter gradient norm at a specified threshold. Constraining the gradient norm is paramount in training models such as recurrent neural networks, which may otherwise suffer from an "exploding gradient" problem (Bengio et al., 1994). Clipping has seen broader value in other neural models, and is abstractly understood as being valuable from an *optimisation* lens: intuitively, by ensuring that the gradient norm is not too large, the dynamics of iterates are more well-behaved (Hazan et al., 2015; Levy, 2016; Zhang et al., 2019). A distinct line of work has also explored the value of gradient clipping from a *privacy* lens (Abadi et al., 2016; Pichapati et al., 2019): intuitively, it prevents any single instance from overly influencing parameter updates, which is a core requirement in ensuring differential privacy (Dwork et al., 2006).

In this paper, we propose a new lens with which to study gradient clipping, namely, *robustness*: intuitively, clipping the gradient prevents over-confident descent steps, which is plausibly beneficial in the presence of noise. Given this intuition, our interest is whether gradient clipping can mitigate the problem of *label noise* in classification, which has received significant recent interest (Scott et al., 2013; Natarajan et al., 2013; Menon et al., 2015; Liu & Tao, 2016; Patrini et al., 2017; Ghosh et al., 2017; Han et al., 2018; Zhang & Sabuncu, 2018; Song et al., 2019; Thulasidasan et al., 2019; Charoenphakdee et al., 2019). We study this question, and provide three main contributions:

(a) we show that gradient clipping alone does *not* endow label noise robustness to even simple models. Specifically, we show that under stochastic gradient descent with linear models, gradient clipping is related to optimising a "Huberised" loss (Lemma 1, 2). While such Huberised losses preserve classification calibration (Lemma 3), they are *not* robust to label noise (Proposition 4).

(b) we propose *composite loss-based gradient clipping*, a variant that *does* have label noise robustness. Specifically, for losses comprising a base loss composed with a link function, (e.g., softmax cross-entropy), we only clip the contribution of the *base loss*. The resulting *partially Huberised loss* preserves classification calibration (Lemma 6), while being robust to label noise (Proposition 7).

(c) we empirically verify that on both synthetic and real-world datasets, partially Huberised versions of standard losses (e.g., softmax cross-entropy) perform well in the presence of label noise (§5).

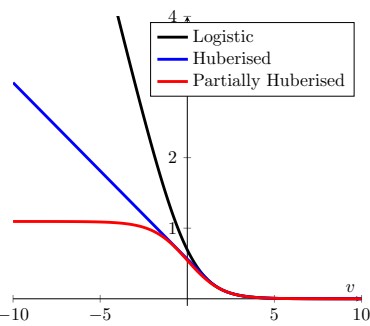 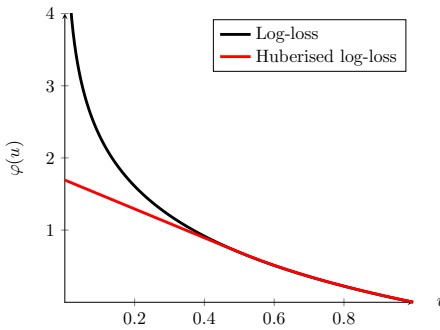

Figure 1: Illustration of the Huberised and partially Huberised logistic loss (left panel). Recall that the logistic loss comprises the log-loss (or cross-entropy) composed with a sigmoid link. The Huberised loss arises from *loss-based gradient clipping*, and linearises the *entire* logistic loss beyond a threshold. The partially Huberised loss arises from *composite loss-based* gradient clipping, and linearises only the *base* loss (i.e., log-loss) beyond a threshold (right panel). Once combined with the sigmoid link, the overall partially Huberised loss asymptotically saturates. Our analysis in §3, §4 establishes that Huberised losses are not robust to label noise, but partially Huberised losses are (Proposition 4, 7).

To illustrate the difference between standard and composite loss-based gradient clipping in (a) and (b), consider the pervasive softmax cross-entropy loss, *viz.* the logistic loss for binary classification. In (a), we relate gradient clipping to loss "Huberisation", which linearises the logistic loss when it exceeds a fixed threshold. In (b), we introduce *composite loss-based* gradient clipping, which is equivalent to a "partial loss Huberisation" that linearises *only* the cross-entropy loss but leaves the sigmoid link untouched. Figure 1 illustrates the Huberised and partially Huberised logistic loss.

## 2 BACKGROUND AND RELATED WORK

We provide background on gradient clipping, loss functions for classification, and label noise.

**Gradient clipping**. Consider a supervised learning task over instances $\mathcal{X}$ and labels $\mathcal{Y}$, where we have a family of models indexed by $\theta \in \Theta$, and the quality of a particular model is measured by a loss function $\ell_\theta \colon \mathcal{X} \times \mathcal{Y} \to \mathbb{R}$. The gradient for a mini-batch $\{(x_n, y_n)\}_{n=1}^N$ is

$$g(\theta) \doteq \frac{1}{N} \sum_{n=1}^N \nabla \ell_\theta(x_n, y_n). \tag{1}$$

One may instead compute the *clipped* gradient, which for user-specified threshold $\tau > 0$ is

$$\bar{g}_\tau(\theta) \doteq \mathtt{clip}_\tau(g(\theta)) \qquad \mathtt{clip}_\tau(w) \doteq \begin{cases} \tau \cdot \frac{w}{||w||_2} & \text{if } ||w||_2 \geq \tau \\ w & \text{else.} \end{cases} \tag{2}$$

Employing $\bar{g}_\tau(\theta)$ for optimisation corresponds to *clipped gradient descent*. This is closely related to normalised gradient descent (NGD) (Hazan et al., 2015), wherein one employs $\tilde{g}(\theta) \doteq \frac{g(\theta)}{||g(\theta)||_2}$. Hazan et al. (2015) showed that NGD can lead to convergence for a wider class of functions than standard gradient descent. Levy (2016) showed that NGD can escape saddle points for non-convex problems. Zhang et al. (2019) showed that gradient clipping can lead to accelerated convergence over gradient descent. Gradient clipping has also been explored for privacy (Abadi et al., 2016; Pichapati et al., 2019), motivated by it preventing any single instance from dominating parameter updates.

**Loss functions**. In binary classification, one observes samples from a distribution $D$ over $\mathcal{X} \times \{\pm 1\}$, and seeks a predictor $f \colon \mathcal{X} \to \mathbb{R}$ with low *risk* $R(f) \doteq \mathbb{E} \ell(y, f(x))$ according to a loss which, in an overload of notation, we denote $\ell \colon \{\pm 1\} \times \mathbb{R} \to \mathbb{R}_+$. For the *zero-one loss* $\ell_{01}(y, f) = [\![ y \cdot f < 0 ]\!]$, $R(f)$ is known as the misclassification risk. Rather than directly using $\ell_{01}$, for computational convenience one often employs a *margin* loss $\ell(y, f) = \phi(y \cdot f)$, for convex $\phi \colon \mathbb{R} \to \mathbb{R}_+$.

We say that $\phi$ is *classification calibrated* (Zhang, 2004; Bartlett et al., 2006) if driving the *excess risk* over the Bayes-optimal predictor for $\phi$ to zero also drives the excess risk for $\ell_{01}$ to zero; that is,

minimising the $\phi$-risk is *statistically consistent* for classification. A canonical example is the hinge loss $\phi(z) = [1 - z]_+$. We call $\phi$ *admissible* if it is "well-behaved" in the sense of being bounded from below, strictly convex, continuously differentiable, non-increasing, and classification calibrated.

We say that $\phi$ is *proper composite* (Reid & Williamson, 2010; Nock & Nielsen, 2009) (or for brevity *composite*) if its minimising scores can be interpreted as probabilities. "Composite" here refers to such losses comprising a *base loss* $\varphi$ composed with an invertible *link function* $F \colon \mathbb{R} \to [0, 1]$. While $\phi$ accepts as input real-valued scores (e.g., the final layer logits of a neural network), these are internally converted to probabilities via $F$. A canonical example is the logistic loss $\phi(z) = -\log F(z)$ with $F \colon z \mapsto \sigma(z)$ for sigmoid $\sigma(z) \doteq (1 + e^{-z})^{-1}$. The multiclass analogue is the softmax cross-entropy loss, wherein the sigmoid becomes a softmax. See Appendix B for a more technical discussion.

**Learning under label noise**. In classification under *label noise*, one has samples from distribution $\bar{D}$ where $\mathbb{P}_{\bar{D}}(y \mid x)$ is a *noisy* version of $\mathbb{P}_D(y \mid x)$, e.g., all labels are corrupted with a fixed constant probability. The goal remains to ensure low risk with respect to the *clean $D$*. This problem has a long history in statistics (Ekholm & Palmgren, 1982; Copas, 1988), and has emerged as a topic of recent interest in machine learning. There has been particular interest in the problem of learning under *symmetric* label noise, wherein all instances have a constant probability of their labels being flipped uniformly to any of the other classes. Scott et al. (2013); Katz-Samuels et al. (2019) proposed a framework for analysing the more general setting of class-conditional label noise. Takenouchi & Eguchi (2004); Natarajan et al. (2013); Cid-Sueiro et al. (2014); Patrini et al. (2017); Natarajan et al. (2018); van Rooyen & Williamson (2018) studied loss-correction for obtaining unbiased risk estimators. Recent works have explored ideas ranging from to model (dis)agreement (Malach & Shalev-Shwartz, 2017; Han et al., 2018; Song et al., 2019) to abstention (Thulasidasan et al., 2019).

## 3 GRADIENT CLIPPING AND HUBERISED LOSSES

We first show that gradient clipping in general *does not* endow robustness to label noise, even in simple settings. Specifically, we establish that stochastic gradient clipping with linear models is equivalent to modifying the underlying loss (Lemma 1). This modified loss is closely related to a *Huberised* loss (Lemma 2), which is equivalent to *loss-based gradient clipping* (*L-gradient clipping*). Unfortunately, Huberised losses, and thus L-gradient clipping, are not robust to label noise (Proposition 4).

### 3.1 FROM GRADIENT TO L-GRADIENT CLIPPING

Consider a binary classification problem, with $\mathcal{Y} = \{\pm 1\}$. Suppose we use a scoring model $s_\theta(x)$ for $\theta \in \Theta$ with *margin* $m_\theta(x, y) \doteq y \cdot s_\theta(x)$, and margin loss $\ell_\theta(x, y) \doteq \phi(m_\theta(x, y))$ for admissible $\phi \colon \mathbb{R} \to \mathbb{R}_+$. Then,

$$\nabla_\theta \ell_\theta(x, y) = \nabla_\theta m_\theta(x, y) \cdot \phi'(m_\theta(x, y)). \tag{3}$$

Now suppose we perform stochastic gradient descent (i.e., we use $N = 1$ in (1)) and use a linear scorer $s_\theta(x) = \theta^{\mathrm{T}} x$. Here, gradient clipping of (3) is equivalent to modifying the loss as follows.

**Lemma 1.** *Pick any admissible margin loss $\phi$, and $\tau > 0$. Then, for loss $\ell_\theta(x, y) \doteq \phi(m_\theta(x, y))$ with linear scorer, the clipped loss gradient (2) is equivalent to the gradient of a modified loss $\bar{\ell}$:*

$$\mathtt{clip}_\tau \left( \nabla \ell_\theta(x, y) \right) = \nabla \bar{\ell}_\theta(x, y),$$

$$\text{with } \bar{\ell}_\theta(x, y) \doteq \begin{cases} -\frac{\tau}{\|\nabla m_\theta(x, y)\|_2} \cdot m_\theta(x, y) & \text{if } |\phi'(m_\theta(x, y))| \geq \frac{\tau}{\|\nabla m_\theta(x, y)\|_2} \\ \phi(m_\theta(x, y)) & \text{else.} \end{cases} \tag{4}$$

The loss $\bar{\ell}$ in (4) is intuitive: if the derivative of the original loss $\phi$ exceeds a certain *effective clipping threshold*, we replace $\phi$ with a linear loss function. This effective threshold $\tau \cdot (\|\nabla m_\theta(x, y)\|_2)^{-1}$ takes into account the *instance-dependent* margin gradient, *viz.* $\|x\|_2$ for linear $s_\theta(x)$.

We will comment in §4.4 as to the properties of gradient clipping in more general scenarios. For the moment, we note that Lemma 1 is closely related to the "Huber loss" (Huber, 1964),

$$\ell_{\mathrm{huber}}(x, y; \theta) = \begin{cases} \tau \cdot |s_\theta(x) - y| & \text{if } |s_\theta(x) - y| \geq \tau \\ \frac{1}{2} \cdot (s_\theta(x) - y)^2 & \text{else.} \end{cases} \tag{5}$$

This replaces the extremes of the square loss with the absolute loss. Both the absolute and Huber losses are widely employed in robust regression (Huber, 1981, Chapter 3), (Hampel et al., 1986, pg. 100). Note that (4) slightly differs from (5) as the effective clipping threshold is instance-dependent. One may nonetheless arrive at Huber-style losses via a variant of gradient clipping; this connection will prove useful for our subsequent analysis. Per (3), gradient clipping involves using

$$\texttt{clip}_\tau\left(\nabla\ell_\theta(x,y)\right) = \texttt{clip}_\tau\left(\nabla m_\theta(x,y)\cdot\phi'(m_\theta(x,y))\right), \tag{6}$$

wherein the clipping considers the margin gradient. Consider now the following *loss-based* gradient clipping (*L-gradient clipping*), wherein we clip *only* the contribution arising from the loss:

$$\texttt{l-clip}_\tau\left(\nabla\ell_\theta(x,y)\right) \doteq \nabla m_\theta(x,y)\cdot\texttt{clip}_\tau\left(\phi'(m_\theta(x,y))\right). \tag{7}$$

Compared to (6), we effectively treat the margin gradient norm $\|\nabla m_\theta(x,y)\|_2$ as constant across instances, and focus on bounding the loss derivative. The latter may be a significant component in the gradient norm; e.g., for linear models, $\|\nabla m_\theta(x,y)\|_2 = \|x\|_2$ is often bounded. Observe further that for linear models with $\|x\|_2 \equiv R$ across instances, (7) is a rescaled version of the clipped gradient:

$$\texttt{clip}_\tau\left(\nabla\ell_\theta(x,y)\right) = \tau\cdot\frac{y\cdot x\cdot\phi'(m_\theta(x,y))}{\tau\vee(R\cdot|\phi'(m_\theta(x,y))|)} = \nabla m_\theta(x,y)\cdot\texttt{clip}_{\tau/R}\left(\phi'(m_\theta(x,y))\right).$$

Interestingly, L-gradient clipping equivalently uses the following "Huberised" version of the loss.

**Lemma 2.** *Pick any admissible margin loss* $\phi\colon\mathbb{R}\to\mathbb{R}_+$ *with Fenchel conjugate* $\phi^*$, *and* $\tau > 0$. *Then, the clipped gradient in* (7) *is equivalent to employing a* Huberised *loss function* $\bar{\phi}_\tau$ *such that*

$$\texttt{clip}_\tau\left(\phi'(z)\right) = \bar{\phi}'_\tau(z) \text{ with } \bar{\phi}_\tau(z) \doteq \begin{cases} -\tau\cdot z - \phi^*(-\tau) & \textit{if } \phi'(z) \leq -\tau \\ \phi(z) & \textit{else.} \end{cases} \tag{8}$$

Evidently, $\bar{\phi}_\tau$ linearises $\phi$ once its derivative is sufficiently large, akin to the Huber loss in (5). One may verify that for $\phi(z) = (1-z)^2$, one arrives exactly at (5).

> **Example 3.1:** For the logistic loss $\phi(z) = \log(1 + e^{-z})$, the Huberised loss for $\tau \in (0,1)$ is
>
> $$\bar{\phi}_\tau(z) = \begin{cases} -\tau\cdot z - \log(1-\tau) - \tau\cdot\sigma^{-1}(\tau) & \text{if } z \leq -\sigma^{-1}(\tau) \\ \log(1 + e^{-z}) & \text{else.} \end{cases}$$
>
> Per Figure 1a, this linearises $\phi$ beyond a fixed threshold. See Appendix C for further illustrations.

The use of Huberised losses in classification is not new (Zhang et al., 2002; Zhang & Johnson, 2003; Rosset & Zhu, 2007). However, we now provide a novel study of their label noise robustness.

## 3.2 CONSISTENCY AND LABEL-NOISE ROBUSTNESS OF HUBERISED LOSSES

Before studying the effect of label noise on Huberised losses, it is apposite to consider whether they are suitable for use even in the absence of noise. One way to formalise this is to ask whether the losses maintain *classification calibration*, in the sense of §2. This is a minimal requirement on a loss to be useful for classification (Zhang, 2004; Bartlett et al., 2006). One may further ask whether the losses preserve class-probabilities, i.e., are *proper composite* in the sense of §2. It is desirable to preserve this key trait of losses such as the logistic loss. The following clarifies both points.

**Lemma 3.** *Pick any admissible margin loss* $\phi$ *and* $\tau > 0$. *Then, the Huberised loss* $\bar{\phi}_\tau$ *in* (8) *is classification calibrated. If* $\phi$ *is proper composite and* $\tau \geq -\phi'(0)$, *then* $\bar{\phi}_\tau$ *is also proper composite.*

L-gradient clipping is thus benign for classification, generalising Rosset & Zhu (2007, Section 3.4), which was for the square-hinge loss. Interestingly, for composite $\phi$ and small $\tau$, the proof reveals that $\bar{\phi}_\tau$ has a non-invertible link function. Intuitively, such $\bar{\phi}_\tau$ are effectively linear, and linear losses are unsuited for estimating probabilities (van Rooyen et al., 2015; Charoenphakdee et al., 2019).

We now turn to our central object of inquiry: does gradient clipping endow robustness to label noise? To study this, we consider L-gradient clipping, which as noted above is a special case of gradient clipping under linear models with constant $\|\nabla m_\theta(x,y)\|_2$. Since L-gradient clipping is in turn equivalent to using a Huberised loss, we may study the robustness properties of this loss.

Surprisingly, when our loss is convex (e.g., softmax cross-entropy), Huberised losses are *not* robust to even very simple forms of label noise. Essentially, since these losses are still convex, they can still be affected by errant outlier observations. Formally, using a result of Long & Servedio (2010) (see Appendix E.1), we arrive at the following.

**Proposition 4.** *Pick any admissible margin loss $\phi$ and $\tau > 0$. Then, $\exists$ a separable distribution for which the optimal linear classifier under $\bar{\phi}_\tau$ is equivalent to random guessing under symmetric noise.*

To situate Proposition 4 in a broader context, we note that for *regression* problems, it is well known that the Huber loss is susceptible to "high leverage" outliers (Hampel et al., 1986, pg. 313), (Rousseeuw & Leroy, 1987, pg. 13), (Maronna et al., 2019, pg. 104), i.e., extremal instances which dominate the optimal solution. Proposition 4 complements these for the case of label noise in classification.

Given that gradient clipping does not endow label noise robustness, how else might we proceed? In a regression context, the outlier vulnerability of the Huber loss can be addressed by using a *trimmed average* of the loss (Rousseeuw & Leroy, 1987; Bhatia et al., 2015; Yang et al., 2018). Such ideas have been successfully explored for label noise problems (Shen & Sanghavi, 2019). We will however demonstrate that a simple variant of clipping yields a loss that *does* possess label noise robustness.

## 4 COMPOSITE LOSS-BASED GRADIENT CLIPPING

We now show that noise robustness can be achieved with *CL-gradient clipping*, a variant wherein for composite losses (e.g., softmax cross-entropy), we perform *partial Huberisation* of the base loss *only*.

### 4.1 FROM HUBERISATION TO PARTIAL HUBERISATION

Consider a *composite* margin loss $\ell_\theta(x, y) \doteq \phi(m_\theta(x, y))$, where $\phi = \varphi \circ F$ for some base loss $\varphi$ and invertible link $F \colon \bar{\mathbb{R}} \to [0, 1]$; e.g., the logistic loss has $\varphi(u) = -\log u$ and $F(z) = \sigma(z)$. We can interpret $p_\theta(x, y) \doteq F(m_\theta(x, y))$ as a probability estimate; e.g., $p_\theta(x, 1) = \sigma(m_\theta(x, 1))$ is the probability of $x$ being positive. Now, rewriting $\ell_\theta(x, y) = \varphi(p_\theta(x, y))$, we may express (3) as:

$$\nabla \ell_\theta(x, y) = \nabla p_\theta(x, y) \cdot \varphi'(p_\theta(x, y)).$$

L-gradient clipping in (7) was defined as only clipping $\phi' = (\varphi \circ F)'$ above, which ensures that the resulting Huberised loss is Lipschitz. Typically, however, $F$ is already Lipschitz; e.g., this is the case for the commonly used sigmoid or softmax link. This suggests the following *composite loss-based gradient clipping* (*CL-gradient clipping*), wherein we *only* clip the derivative for the *base* loss $\varphi$:

$$\texttt{cl-clip}_\tau \left( \nabla \ell_\theta(x, y) \right) \doteq \nabla p_\theta(x, y) \cdot \texttt{clip}_\tau \left( \varphi'(p_\theta(x, y)) \right). \tag{9}$$

As before, CL-gradient clipping corresponds to optimising a new, "partially Huberised" loss.

**Lemma 5.** *Pick any admissible, composite margin loss $\phi = \varphi \circ F$, and $\tau > 0$. Then, the clipped gradient in (9) is equivalent to employing a* partially Huberised loss $\tilde{\phi}_\tau = \tilde{\varphi}_\tau \circ F$, where

$$\tilde{\varphi}_\tau(u) \doteq \begin{cases} -\tau \cdot u - \varphi^*(-\tau) & \text{if } \varphi'(u) \leq -\tau \\ \varphi(u) & \text{else.} \end{cases} \tag{10}$$

Compared to the Huberised loss (7), the partially Huberised loss only linearises the *base* loss $\varphi$, while retaining the link $F$. Consequently, the *composite* loss $\tilde{\phi}_\tau$ behaves like the link beyond a certain threshold, and will thus be bounded: indeed, $\tilde{\phi}_\tau(z) = -\tau \cdot F(z) - \varphi^*(-\tau)$ if $\varphi'(F(z)) \leq -\tau$.

> **Example 4.1:** For the logistic loss, $\phi(z) = \varphi(F(z))$ with $\varphi(u) = -\log u$ and $F(z) = \sigma(z)$,
>
> $$\tilde{\phi}_\tau(z) = \begin{cases} -\tau \cdot \sigma(z) + \log \tau + 1 & \text{if } z \leq \sigma^{-1}\left(\frac{1}{\tau}\right) \\ \log(1 + e^{-z}) & \text{else.} \end{cases} \tag{11}$$

Note that partial Huberisation readily generalises to a multi-class setting. Indeed, suppose we have softmax probability estimates $p_\theta(x, y) \propto \exp(m_\theta(x, y))$. Then, whereas the softmax cross-entropy employs $\ell_\theta(x, y) = -\log p_\theta(x, y)$, our *partially Huberised softmax cross-entropy* for $\tau > 1$ is

$$\tilde{\ell}_\theta(x, y) = \begin{cases} -\tau \cdot p_\theta(x, y) + \log \tau + 1 & \text{if } p_\theta(x, y) \leq \frac{1}{\tau} \\ -\log p_\theta(x, y) & \text{else.} \end{cases} \tag{12}$$

### 4.2 CONSISTENCY AND LABEL NOISE ROBUSTNESS OF PARTIALLY HUBERISED LOSSES

Following §3.2, we establish that CL-gradient clipping is *always* benign from a classification perspective, and provided $\tau$ is sufficiently large, from a probability estimation perspective as well. As before, we do this by exploiting the equivalence of CL-gradient clipping to a partially Huberised loss.

**Lemma 6.** *Pick any admissible composite margin loss $\phi = \varphi \circ F$ and $\tau > 0$. Then, the loss $\tilde{\phi}_\tau$ in (10) is classification calibrated. If further $\tau \geq -\varphi'(1/2)$, then $\tilde{\varphi}_\tau$ is also proper composite.*

We now show that partially Huberised losses have an important advantage over Huberised losses: under symmetric label noise, the optimal solution on the *clean* distribution cannot be *too* far away from the optimal solution on the *noisy* distribution. This implies that label noise (such as that considered in Proposition 4) cannot have an excessively deleterious influence on the loss.

**Proposition 7.** *Pick any proper loss $\varphi$ and $\tau > 0$. Let $f^*$ be the risk minimiser of $\tilde{\varphi}_\tau$ on the* clean *distribution. For any non-trivial level of symmetric label noise, let $\overline{\text{reg}}_\tau(f^*)$ denote the excess risk of $f^*$ with respect to $\tilde{\varphi}_\tau$ on the* noisy *distribution. Then, there exists $C > 0$ such that $\overline{\text{reg}}_\tau(f^*) \leq C$.*

Note that by van Rooyen et al. (2015, Proposition 4), it is impossible for the above bound to hold with $C = 0$ without using a linear loss. Nonetheless, by virtue of partially Huberised losses being *partially* linear, we are able to bound the degradation under label corruption. The saturating behaviour of the partially Huberised loss also implies robustness to outliers in feature space; see Appendix D.

### 4.3 RELATION TO EXISTING WORK

The partially Huberised log loss in (10), (11) can be related to a family of losses studied in several works (Ding & Vishwanathan, 2010; Hung et al., 2018; Zhang & Sabuncu, 2018; Sypherd et al., 2019; Amid et al., 2019b;a): for $\alpha \in (0, 1]$, define the *generalised cross-entropy* $\varphi_\alpha \colon [0, 1] \to \bar{\mathbb{R}}_+$ by $\varphi_\alpha(u) \doteq (1 - u^\alpha)/\alpha$. See Appendix C.3 for an illustration of $\varphi_\alpha$. There are two similarities between (11) and $\varphi_\alpha$. First, both proposals interpolate between the log and linear losses: when $\alpha \to 0^+$, $\varphi_\alpha$ approaches the log loss, and when $\alpha = 1$, $\varphi_\alpha$ equals the linear loss. Second, both proposals modify the base loss, allowing the link $F$ to be chosen independently. In particular, one may use the heavy-tailed $F$ of Ding & Vishwanathan (2010); Amid et al. (2019b) in conjunction with our partially Huberised loss.

One difference between (11) and $\varphi_\alpha$ is that the partially Huberised loss is *exactly* linear for a suitable region; consequently, it is guaranteed to be Lipschitz, unlike $\varphi_\alpha$. This can be understood in terms of the loss gradients: for a class-probability estimate $p_\theta(x, y)$, let $\ell_\theta(x, y) \doteq \varphi(p_\theta(x, y))$. For the generalised cross-entropy, $\nabla \ell_\theta(x, y) = -p_\theta(x, y)^{\alpha-1} \cdot \nabla p_\theta(x, y)$, while for the partially Huberised loss, $\nabla \ell_\theta(x, y) = -(p_\theta(x, y) \vee \tau)^{-1} \cdot \nabla p_\theta(x, y)$. Both gradients thus take into account whether a sample is "informative", in the sense of being poorly-predicted ($p_\theta(x, y) \sim 0$). Further, to guard against such samples being the result of label noise, both ensure this influence is not overwhelming, but in different ways: the generalised cross-entropy softens the influence, while still allowing it to be unbounded as $p_\theta(x, y) \to 0$. On the other hand, the partially Huberised loss enforces a hard cap of $\tau^{-1}$ on the influence. This is to be contrast with a *truncated* loss also considered in Zhang & Sabuncu (2018), which enforces a hard cap on the *loss*, thus completely discarding poorly-predicted instances.

As shown in Figure 1a, the partially Huberised loss in (10) exhibits saturating behaviour at the extremes. This by itself is hardly new in the context of robust statistics. For example, Tukey's "biweight" loss replaces the Huber loss' linear behaviour with hard saturation (Gross & Tukey, 1973), and is a member of the broader family of *redescending M-estimators* (Andrews et al., 1972; Shevlyakov et al., 2008). In a similar spirit, Masnadi-Shirazi & Vasconcelos (2009) proposed the bounded Savage loss $\varphi(v) = (\sigma(2v) - 1)^2$, while Yang et al. (2010); Menon & Williamson (2018) theoretically studied loss clipping. Robust versions of logistic regression derived by variants of the Huber loss have been explored, e.g., Pregibon (1982, Section 4.1), Copas (1988, Section 3.1), (Feng et al., 2014).

The distinction of the above proposals to our work is two-fold. First, we explicitly consider the classification setting, wherein we show consistency and robustness to label noise for the partially Huberised loss. Second, our proposal explicitly highlights the connection to gradient clipping.

Finally, while our focus has been on label noise, there are other notions of "robustness". Indeed, several works have proposed means of making gradient descent robust to heavy-tailed distributions

| Clipping type | Clipped gradient form | Equivalent loss | Reference | Label noise robust? |
|---|---|---|---|---|
| Gradient | $\mathtt{clip}_\tau\left(\nabla m(\theta) \cdot \phi'(m(\theta))\right)$ | Equation 4 | Equation 6 | $\times$ (Proposition 4) |
| L-Gradient | $\nabla m(\theta) \cdot \mathtt{clip}_\tau\left(\phi'(m(\theta))\right)$ | Huberised | Equation 7 | $\times$ (Proposition 4) |
| CL-Gradient | $\nabla p(\theta) \cdot \mathtt{clip}_\tau\left(\varphi'(p(\theta))\right)$ | Partially Huberised | Equation 9 | $\checkmark$ (Proposition 7) |

Table 1: Summary of types of gradient clipping considered in this paper. We consider binary classification problems involving a labelled example $(x, y)$, parametrised scoring function $s_\theta(x)$ with margin $m(\theta) \doteq y \cdot s_\theta(x)$, and differentiable *composite margin loss* $\phi(z)$. This loss internally converts scores to probabilities $p(\theta) \doteq F(m(\theta))$ for link function $F(\cdot)$, which is evaluated with some *base loss* $\varphi$; i.e., $\phi = \varphi \circ F$. Gradient clipping applies to the full loss, i.e., $\phi(m(\theta))$. L-gradient clipping applies only to the composite loss, leaving the score untouched; this is equivalent to using a Huberised loss. CL-gradient clipping applies only to the base loss, leaving the link untouched; this is equivalent to using a partially Huberised loss. Only the latter has robustness guarantee under symmetric label noise.

or outliers (Brownlees et al., 2015; Hsu & Sabato, 2016; Lugosi & Mendelson, 2016; Lecué et al., 2018; Prasad et al., 2018; Holland & Ikeda, 2019), typically by passing the gradient through a robust operator (Tukey, 1960; Hampel et al., 1986; Rousseeuw & Leroy, 1987; Lerasle & Oliveira, 2011; Catoni, 2012; Minsker, 2015). By contrast, we analyse the (simpler) gradient clipping approach, showing that while it is inadequate to cope with label noise, a simple modification rectifies this. Since our proposal only modifies the underlying loss, it may be combined with any of these methods.

## 4.4 Discussion and implications

Table 1 summarises our results, highlighting the perspective of gradient clipping as equivalently modifying the loss. Before proceeding, we make some qualifying comments. First, our analysis has assumed *symmetric* label noise. Often, one encounters *asymmetric* or *instance-dependent* noise (Menon et al., 2018). While corresponding guarantees for the linear loss may be ported over to the partially Huberised loss, they require stronger distributional assumptions (Ghosh et al., 2015).

Second, Proposition 4 exhibits a *specific* distribution which defeats the Huberised loss under linear models. In practice, distributions may be more benign (Patrini et al., 2016), and models are often nonlinear, meaning that Huberised losses (and thus gradient clipping) are thus unlikely to succumb as extremely to label noise as Proposition 4 suggests. The aim of §4 is however to establish that a simple modification of clipping avoids worst-case degradation, without adding significant complexity.

Third, for minibatch size $N > 1$, the effect of clipping is not a simple loss modification, since the loss gradients for each sample will be modified by the *entire* minibatch loss gradient norm $\|g(\theta)\|_2$. Since this minibatch is randomly drawn, one cannot mimic gradient clipping by a simple modification of the loss function. However, the results for $N = 1$ suffice to establish that gradient clipping cannot *in general* endow robustness. One may use our partially Huberised loss in conjunction with minibatch gradient clipping, to potentially obtain both robustness *and* optimisation benefits.

Finally, partially Huberised losses such as (12) require setting a hyperparameter $\tau$ (e.g., by cross-validation), similar to $\alpha$ in the generalised cross-entropy per §4.3. Intuitively, the optimal $\tau$ trades off the noise-robustness of the linear loss, and the gradient informativeness of the base loss (per the discussion in §4.3). Setting $\tau$ to be large tacitly assumes that one's samples are largely noise-free.

## 5 Experimental illustration

We now present experiments illustrating that: (a) we may exhibit label noise scenarios that defeat a Huberised but not partially Huberised loss, confirming Propositions 4, 7, and (b) partially Huberised versions of existing losses perform well on real-world datasets subject to label noise.

**Synthetic datasets**. Our first experiments involve two synthetic datasets, which control for confounding factors. We begin with a setting from Long & Servedio (2010), comprising a 2D linearly separable distribution. (See Appendix E.1 for an illustration.) We draw $N = 1,000$ random samples from this distribution, and flip each label with $\rho = 45\%$ probability. We train a linear classifier to minimise one of several losses, and evaluate the classifier's accuracy on $500$ *clean* test samples.

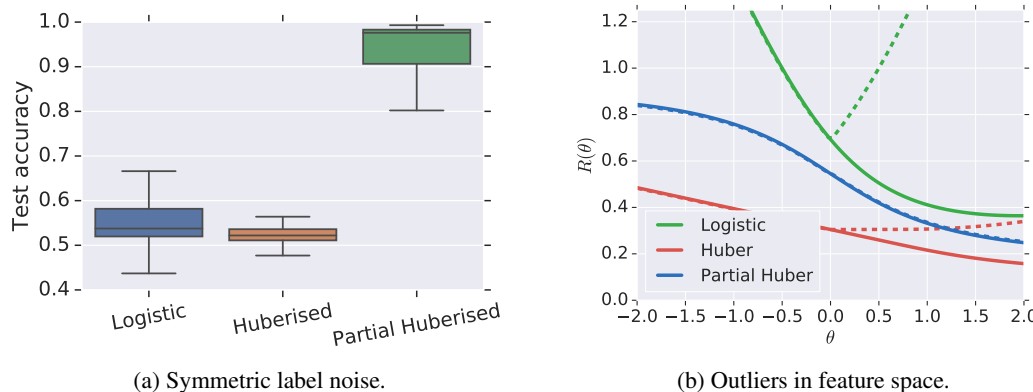

(a) Symmetric label noise.

(b) Outliers in feature space.

Figure 2: Results on synthetic data. In (b), the solid (dashed) curve denotes empirical risk without (with) outliers.

| Dataset | Loss function | $\rho = 0.0$ | $\rho = 0.2$ | $\rho = 0.4$ | $\rho = 0.6$ |
|---|---|---|---|---|---|
| MNIST | CE | $98.7 \pm 0.0$ | $97.7 \pm 0.2$ | $96.4 \pm 0.3$ | $91.1 \pm 0.6$ |
| | CE + clipping | $98.6 \pm 0.1$ | $97.5 \pm 0.1$ | $96.8 \pm 0.1$ | $88.9 \pm 0.8$ |
| | Linear | $98.3 \pm 0.0$ | $88.2 \pm 0.1$ | $78.8 \pm 0.0$ | $9.6 \pm 0.1$ |
| | GCE | $98.8 \pm 0.1$ | $98.6 \pm 0.1$ | $98.2 \pm 0.0$ | $97.6 \pm 0.1$ |
| | PHuber-CE $\tau = 10$ | $98.5 \pm 0.0$ | $98.7 \pm 0.0$ | $98.5 \pm 0.1$ | $97.6 \pm 0.0$ |
| | PHuber-GCE $\tau = 10$ | $98.6 \pm 0.1$ | $98.5 \pm 0.1$ | $98.4 \pm 0.1$ | $97.8 \pm 0.0$ |
| CIFAR-10 | CE | $92.4 \pm 0.1$ | $72.4 \pm 0.2$ | $61.4 \pm 0.8$ | $40.6 \pm 0.3$ |
| | CE + clipping | $88.3 \pm 0.3$ | $68.7 \pm 0.2$ | $56.7 \pm 0.6$ | $44.2 \pm 2.1$ |
| | Linear | $91.3 \pm 0.1$ | $88.2 \pm 0.1$ | $83.8 \pm 0.2$ | $69.3 \pm 0.2$ |
| | GCE | $91.7 \pm 0.0$ | $88.4 \pm 0.2$ | $80.8 \pm 0.3$ | $60.0 \pm 0.3$ |
| | PHuber-CE $\tau = 2$ | $91.6 \pm 0.1$ | $88.6 \pm 0.0$ | $83.6 \pm 0.1$ | $72.2 \pm 0.0$ |
| | PHuber-GCE $\tau = 10$ | $92.0 \pm 0.1$ | $88.5 \pm 0.1$ | $80.8 \pm 0.1$ | $62.6 \pm 0.2$ |
| CIFAR-100 | CE | $66.6 \pm 1.4$ | $49.7 \pm 0.3$ | $29.9 \pm 0.9$ | $11.4 \pm 0.2$ |
| | CE + clipping | $28.8 \pm 0.1$ | $20.6 \pm 0.4$ | $14.7 \pm 0.6$ | $9.0 \pm 0.4$ |
| | Linear | $12.1 \pm 1.6$ | $6.6 \pm 1.2$ | $5.7 \pm 0.9$ | $3.6 \pm 0.1$ |
| | GCE | $70.1 \pm 0.1$ | $63.9 \pm 0.1$ | $52.0 \pm 0.2$ | $29.9 \pm 0.5$ |
| | PHuber-CE $\tau = 10$ | $66.2 \pm 1.5$ | $56.2 \pm 2.2$ | $44.4 \pm 0.7$ | $18.5 \pm 0.4$ |
| | PHuber-GCE $\tau = 10$ | $69.8 \pm 0.2$ | $64.4 \pm 0.4$ | $52.4 \pm 0.2$ | $31.5 \pm 0.8$ |

Table 2: Test set accuracy where the training labels are corrupted with probability $\rho$. We report the mean and standard error over 3 trials. The highlighted cells are the best performing loss at a given $\rho$. "PHuber" here refers to our partial Huberisation from §4, which is equivalent to a variant of gradient clipping.

We compare the logistic loss with its Huberised and partially Huberised versions, using $\tau = 1$ and $\tau = 2$ respectively. Figure 2a presents the results over 500 independent trials. The logistic loss and its Huberised counterpart suffer significantly under noise, while the partially Huberised loss often achieves perfect near-discrimination. This confirms that in the worst case, L-gradient clipping may succumb to noise, while CL-gradient clipping performs well in the same scenario.

We next consider a 1D setting based on Ding (2013, Section 3.2.3), comprising $10,000$ linearly separable "inliers" and $50$ "outliers". Assuming the use of a linear model parameterised by scalar $\theta \in \mathbb{R}$, we plot the empirical risk of the samples with and without the outlier observations as $\theta$ is varied. Figure 2b shows that the logistic loss and its Huberised variant are strongly affected by the outliers: their optimal solution goes from $\theta^* = +\infty$ to $\theta^* = 0$. However, the partially Huberised loss is largely immune to the outliers. Appendix E contains additional details on this experiment.

**Real-world datasets**. We now demonstrate that partially Huberised losses perform well with deep neural networks trained on MNIST, CIFAR-10 and CIFAR-100 (Krizhevsky & Hinton, 2009). For MNIST, we train a LeNet (Lecun et al., 1998) using Adam with batch size $N = 32$, and weight decay

of $10^{-3}$. For CIFAR-10 and CIFAR-100, we train a ResNet-50 (He et al., 2016) using SGD with momentum 0.1, weight decay of $5 \times 10^{-3}$, batch normalisation, and $N = 64, 128$ respectively.

For each dataset, we corrupt the training labels with symmetric noise at flip probability $\rho \in \{0.0, 0.2, 0.4, 0.6\}$. We compare the test set accuracy of various losses combined with a softmax link. Our baseline is the cross-entropy loss (CE). As representative noise-robust losses, we consider the linear or unhinged loss (van Rooyen et al., 2015; Ghosh et al., 2017), and the generalised cross-entropy (GCE) with $\alpha = 0.7$, following Zhang & Sabuncu (2018). We additionally assess global gradient clipping (with $\tau = 0.1$) of the CE, which per §3 is akin to a Huberised loss.

We apply our partial Huberisation of (10) to the CE (12) ("PHuber-CE"), *and* the GCE ("PHuber-GCE"). The latter highlights that partial Huberisation is not tied to the cross-entropy, and is applicable even on top of existing noise-robust losses. Recall that partial Huberisation offers a choice of tuning parameter $\tau$, similar to the $\alpha$ parameter in GCE, and noise-rate estimates in loss-correction techniques more generally. For each dataset, we pick $\tau \in \{2, 10\}$ (equivalently corresponding to probability thresholds 0.5 and 0.1 respectively) so as to maximise accuracy on a validation set of noisy samples with the maximal noise rate $\rho = 0.6$; the chosen value of $\tau$ was then used for each noise level. Tuning $\tau$ separately for each setting of the noise rate $\rho$ can be expected to help performance, at the expense of increased computational cost. Recall also that as $\tau \to 1$, partial Huberisation mimics using the base loss, while as $\tau \to +\infty$, partial Huberisation mimics using the linear loss; our hypothesis is that an intermediate $\tau$ can attain a suitable balance between noise robustness, and gradient informativeness.

Table 2 shows that in the noise-free case ($\rho = 0.0$), all methods perform comparably. However, when injecting noise, accuracy for the CE degrades dramatically. Further, gradient clipping sometimes offers improvements under high noise; however, the performance is far inferior to other losses, which is in keeping with their robustness guarantees. Indeed, the linear loss, which is provably robust to symmetric noise, generally performs well even when $\rho = 0.6$. However, optimisation under this loss is more challenging, since the gradient does not account for instances' importance (per §4.3). This is particularly reflected on the CIFAR-100 dataset, where this loss suffers to learn even under no noise. The GCE and partially Huberised losses do not suffer from this issue, even at high noise levels.

Generally, the partially Huberised losses are competitive with or improve upon the counterpart losses they build upon. In particular, the partially Huberised CE performs much better than the CE under high noise, while the partially Huberised GCE slightly bumps up the GCE numbers on CIFAR-100. This indicates that partially Huberised may be useful in combining with generic base losses to cope with noise. We reiterate here that our partially Huberised loss may be used in conjunction with other ideas, e.g., pruning (Zhang & Sabuncu, 2018), consensus (Malach & Shalev-Shwartz, 2017; Han et al., 2018), or abstention (Thulasidasan et al., 2019). We leave such exploration for future work.

## 6    CONCLUSION AND FUTURE WORK

We established that gradient clipping by itself does *not* suffice to endow even simple models with label noise robustness; however, a simple variant resolves this issue. Experiments confirm that our composite loss-based gradient clipping performs well on datasets corrupted with label noise. One interesting direction for future work is to analyse the behaviour of gradient-clipping inspired losses for the more general problem of distributionally robust learning (Shafieezadeh-Abadeh et al., 2015; Namkoong & Duchi, 2016; 2017; Hu et al., 2018; Sinha et al., 2018; Duchi & Namkoong, 2019).

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

## A    PROOFS OF RESULTS

*Proof of Lemma 1.* Following Equation 3,

$$\nabla \ell_\theta(x, y) = \nabla m_\theta(x, y) \cdot \phi'(m_\theta(x, y)).$$

Similarly, for the linear loss function $\ell_{\mathrm{lin}}(x, y; \theta) \doteq -m_\theta(x, y)$, we have

$$\nabla \ell_{\mathrm{lin}}(x, y; \theta) = -\nabla m_\theta(x, y).$$

To compute the normalised gradient, we need

$$\|\nabla \ell_\theta(x, y)\|_2 = \|\nabla m_\theta(x, y)\|_2 \cdot |\phi'(m_\theta(x, y))|.$$

Thus,

$$
\begin{aligned}
\frac{\nabla \ell_\theta(x, y)}{\|\nabla \ell_\theta(x, y)\|_2} &= \frac{1}{\|\nabla m_\theta(x, y)\|_2} \cdot \nabla m_\theta(x, y) \cdot \frac{\phi'(m_\theta(x, y))}{|\phi'(m_\theta(x, y))|} \\
&= \frac{1}{\|\nabla m_\theta(x, y)\|_2} \cdot \nabla m_\theta(x, y) \cdot \mathrm{sign}(\phi'(m_\theta(x, y))) \\
&= \frac{1}{\|\nabla m_\theta(x, y)\|_2} \cdot \nabla \ell_{\mathrm{lin}}(x, y; \theta),
\end{aligned}
$$

since $\phi'(z) < 0$ by assumption that $\phi$ is admissible, and thus decreasing. Consequently, if $N = 1$,

$$
\bar{g}(\theta) = \mathrm{clip}_\tau(\nabla \ell_\theta(x, y)) = \begin{cases} \frac{\tau}{\|\nabla m_\theta(x, y)\|_2} \cdot \nabla \ell_{\mathrm{lin}}(x, y; \theta) & \text{if } |\phi'(m_\theta(x, y))| \ge \frac{\tau}{\|\nabla m_\theta(x, y)\|_2} \\ \nabla \ell_\theta(x, y) & \text{else.} \end{cases}
$$

Assuming a linear scorer $s(x; \theta) = \theta^{\mathrm{T}} x$, the score gradient $\nabla s(x; \theta) = x$, which is independent of $\theta$. The clipped gradient thus corresponds to the gradient under a "Huberised" loss function

$$
\bar{\ell}_\theta(x, y) = \begin{cases} \frac{\tau}{\|\nabla m_\theta(x, y)\|_2} \cdot \ell_{\mathrm{lin}}(x, y; \theta) & \text{if } |\phi'(m_\theta(x, y))| \ge \frac{\tau}{\|\nabla m_\theta(x, y)\|_2} \\ \ell_\theta(x, y) & \text{else.} \end{cases}
$$

$\square$

*Proof of Lemma 2.* Since $\phi$ is strictly convex and decreasing, it must be strictly decreasing. We have

$$
\begin{aligned}
\mathrm{clip}_\tau(\phi'(z)) &= \begin{cases} \tau \cdot \frac{\phi'(z)}{|\phi'(z)|} & \text{if } |\phi'(z)| \ge \tau \\ \phi'(z) & \text{else} \end{cases} \\
&= \begin{cases} \tau \cdot \mathrm{sign}(\phi'(z)) & \text{if } |\phi'(z)| \ge \tau \\ \phi'(z) & \text{else} \end{cases} \\
&= \begin{cases} -\tau & \text{if } \phi'(z) \le -\tau \\ \phi'(z) & \text{else} \end{cases} \quad \text{since } \phi' \text{ is strictly decreasing} \\
&= \begin{cases} -\tau & \text{if } z \le (\phi')^{-1}(-\tau) \\ \phi'(z) & \text{else,} \end{cases}
\end{aligned}
$$

where $(\phi')^{-1}$ exists since $\phi'$ is strictly inceasing by definition of strict convexity.

Now, by definition, the Huberised loss is

$$
\bar{\phi}_\tau(z) = \begin{cases} -\tau \cdot z - \phi^*(-\tau) & \text{if } \phi'(z) \le -\tau \\ \phi(z) & \text{else,} \end{cases}
$$

and so

$$
\bar{\phi}'_\tau(z) = \begin{cases} -\tau & \text{if } \phi'(z) \le -\tau \\ \phi'(z) & \text{else,} \end{cases}
$$

which exactly equals $\mathrm{clip}_\tau(\phi'(z))$. We remark here that $\bar{\phi}_\tau$ is continuous, since for any convex conjugate,

$$\phi^*(-u) = -u \cdot (\phi')^{-1}(-u) - \phi((\phi')^{-1}(-u)).$$

Plugging in $u = \tau$, at the intersection point $z_0 \doteq (\phi)'^{-1}(-\tau)$ of the two pieces of the function,

$$\bar{\phi}_\tau(z_0) = -\tau \cdot (\phi')^{-1}(-\tau) - \phi^*(-\tau) = \phi((\phi')^{-1}(-\tau)) = \phi(z_0).$$

$\square$

*Proof of Lemma 3.* For admissible $\phi$, the Huberised loss $\bar{\phi}_\tau$ of (8) is trivially convex, differentiable everywhere, and decreasing. In particular, we must have $\bar{\phi}'_\tau(0) < 0$, and so $\bar{\phi}_\tau$ must be classification calibrated by Bartlett et al. (2006, Theorem 2). As an illustration, Figure 3 shows the minimiser of the conditional risk,

$$z^*(\eta) \doteq \operatorname*{argmin}_{z \in \mathbb{R}} \left[ \eta \cdot \tilde{\phi}_\tau(z) + (1 - \eta) \cdot \tilde{\phi}_\tau(-z) \right].$$

Note that this quantity must be non-negative if and only if $\eta > 1/2$ for a loss to be classification calibrated. This is easily verified to be true for the Huberised logistic loss, regardless of $\tau$.

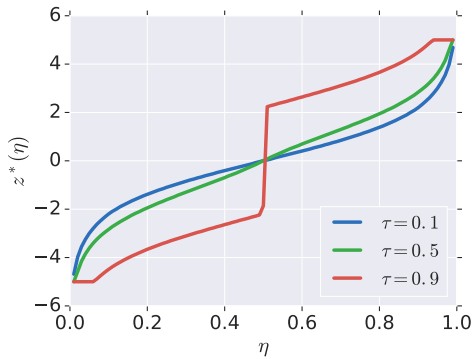

Figure 3: Risk minimiser for the Huberised logistic loss.

Now suppose that $\phi$ is a proper composite loss. To show that $\bar{\phi}_\tau$ is proper composite, it suffices by Williamson et al. (2016, Corollary 12) to show that $\frac{\bar{\phi}'_\tau(z)}{\bar{\phi}'_\tau(-z)}$ is strictly monotone and continuous. Continuity is immediate; for monotonicity, observe that by definition,

$$\bar{\phi}'_\tau(z) = \begin{cases} -\tau & \text{if } z \le (\phi')^{-1}(-\tau) \\ \phi'(z) & \text{else.} \end{cases}$$

For brevity, let $z_0 \doteq (\phi')^{-1}(-\tau)$, where we take $z_0 = -\infty$ if $-\tau \notin \operatorname{ran}(\phi')$. Then,

$$\bar{\phi}'_\tau(z) = \begin{cases} \phi'(z_0) & \text{if } z \le z_0 \\ \phi'(z) & \text{else.} \end{cases}$$

$$\bar{\phi}'_\tau(-z) = \begin{cases} \phi'(z_0) & \text{if } z \ge -z_0 \\ \phi'(-z) & \text{else.} \end{cases}$$

Thus, the quantity of interest is

$$\frac{\bar{\phi}'_\tau(z)}{\bar{\phi}'_\tau(-z)} = \begin{cases} \frac{\phi'(z_0)}{\phi'(-z)} & \text{if } z \le z_0 \\ \frac{\phi'(z)}{\phi'(-z)} & \text{if } z_0 \le z \le -z_0 \\ 1 & \text{if } -z_0 \le z \le z_0 \\ \frac{\phi'(z)}{\phi'(z_0)} & \text{if } z \ge -z_0. \end{cases}$$

Since $\phi$ is strictly convex, $\phi'$ is strictly decreasing and thus invertible. Thus, the ratio $\frac{\bar{\phi}'_\tau(z)}{\bar{\phi}'_\tau(-z)}$ is invertible, provided $z_0 \le 0$, i.e., $\tau \ge -\phi'(0)$. Consequently, $\bar{\phi}_\tau$ is proper composite when $\tau \ge -\phi'(0)$. To intuit the need for the restriction on $\tau$, observe that by (Reid & Williamson, 2010, Corollary 12) the tentative link function for the loss is

$$\bar{F}(z) \doteq \frac{1}{1 - \frac{\bar{\phi}'(z)}{\bar{\phi}'(-z)}}$$

$$= \begin{cases} \frac{\phi'(-z)}{\phi'(z_0) + \phi'(-z)} & \text{if } z \le z_0 \\ F(z) & \text{if } z_0 \le z \le -z_0 \\ \frac{1}{2} & \text{if } -z_0 \le z \le z_0 \\ \frac{\phi'(z_0)}{\phi'(z_0) + \phi'(z)} & \text{if } z \ge -z_0. \end{cases}$$

When $\tau < -\phi'(0)$, the above is seen to be *non-invertible*. See also Appendix C for an illustration of this link function for the logistic loss. □

*Proof of Proposition 4.* In order to apply Long & Servedio (2010, Theorem 2), we simply need to check that the loss $\bar{\phi}_\tau$ is a convex potential in the sense of Long & Servedio (2010, Definition 1). This requires that $\bar{\phi}_\tau$ is convex, non-increasing, continuously differentiable, and asymptotes to zero (or equally, is bounded from below). Each of these is satisfied by assumption of $\phi$ being admissible. □

*Proof of Lemma 5.* By Lemma 2, we may write $\text{clip}_\tau(\varphi'(u))$ as the derivative of a partially Huberised base loss $\tilde{\varphi}_\tau$ given by

$$\tilde{\varphi}_\tau(u) \doteq \begin{cases} -\tau \cdot u - \varphi^*(-\tau) & \text{if } \varphi'(u) \leq -\tau \\ \varphi(u) & \text{else} \end{cases}$$

This induces a composite margin loss $\tilde{\phi}_\tau \doteq \tilde{\varphi}_\tau \circ F$. Now let $p_\theta(x, y) = F(m_\theta(x, y))$, and define $\tilde{\ell}_\tau(x, y; \theta) \doteq \tilde{\phi}_\tau(m_\theta(x, y))$. The gradient under this loss is

$$\begin{aligned}
\nabla_\theta \tilde{\ell}_\tau(x, y; \theta) &= \nabla_\theta \tilde{\phi}_\tau(m_\theta(x, y)) \\
&= \nabla_\theta \tilde{\varphi}_\tau(p_\theta(x, y)) \\
&= \nabla_\theta p_\theta(x, y) \cdot \tilde{\varphi}'_\tau(p_\theta(x, y)) \\
&= \nabla_\theta p_\theta(x, y) \cdot \text{clip}_\tau(\varphi'(p_\theta(x, y))) \\
&= \text{cl} - \text{clip}_\tau(\nabla \ell_\theta(x, y)).
\end{aligned}$$

Thus, CL-gradient clipping is equivalent to using the loss $\tilde{\ell}_\tau$. □

*Proof of Lemma 6.* We proceed in a similar manner to Lemma 3: to show that the loss is proper, we must establish that $\frac{\tilde{\phi}'_\tau(z)}{\tilde{\phi}'_\tau(-z)}$ is invertible. By Reid & Williamson (2010, Corollary 14), for the margin loss $\phi$ to be proper composite with link $F$, it must be true that $F$ satisfies the symmetry condition $F(-z) = 1 - F(z)$. We thus have

$$\begin{aligned}
\tilde{\phi}'_\tau(z) &= F'(z) \cdot \tilde{\varphi}'_\tau(F(z)) \\
\tilde{\phi}'_\tau(-z) &= F'(-z) \cdot \tilde{\varphi}'_\tau(F(-z)) \\
&= -F'(z) \cdot \varphi'(1 - F(z)).
\end{aligned}$$

Consequently,

$$\frac{\tilde{\phi}'_\tau(z)}{\tilde{\phi}'_\tau(-z)} = -\frac{\tilde{\varphi}'_\tau(F(z))}{\tilde{\varphi}'_\tau(1 - F(z))}.$$

Since $F$ is invertible by assumption, the above is invertible if and only if $\frac{\tilde{\varphi}'_\tau(u)}{\tilde{\varphi}'_\tau(1-u)}$ is invertible. Observe that

$$\tilde{\varphi}'_\tau(u) = \begin{cases} -\tau & \text{if } u \leq (\varphi')^{-1}(-\tau) \\ \varphi'(u) & \text{else.} \end{cases}$$

Let $u_0 \doteq (\varphi')^{-1}(-\tau)$ for brevity. Then,

$$\frac{\tilde{\varphi}'_\tau(u)}{\tilde{\varphi}'_\tau(1-u)} = \begin{cases} -\frac{\tau}{\varphi'(1-u)} & \text{if } u \leq u_0 \wedge (1 - u_0) \\ \frac{\varphi'(u)}{\varphi'(1-u)} & \text{if } u_0 \leq u \leq 1 - u_0 \\ 1 & \text{if } 1 - u_0 \leq u \leq u_0 \\ -\frac{\varphi'(u)}{\tau} & \text{if } u \geq u_0 \vee (1 - u_0) \end{cases}$$

This quantity is invertible provided $u_0 \leq \frac{1}{2}$, i.e., $\tau \geq -\varphi'(1/2)$. A subtlety, however, is that the above does not necessarily span the entire range of $[0, +\infty]$; consequently, $\tilde{\varphi}_\tau$ *itself* is proper composite, with a link function of its own.

Even when $\tau$ is small, one may verify that the loss $\tilde{\phi}_\tau$ is nonetheless classification calibrated: this is because for any $\eta \in [0, 1]$, the minimiser $z^*(\eta)$ of the conditional risk

must satisfy the stationarity condition

$$-\frac{\tilde{\phi}'_\tau(z^*)}{\tilde{\phi}'_\tau(-z^*)} = \frac{\eta}{1-\eta}.$$

We thus need to find a suitable $z^*$ such that the left hand side equates to a given constant. Now, for any $C \neq 1$ there is a unique $u$ such that $\frac{\tilde{\varphi}'_\tau(u)}{\tilde{\varphi}'_\tau(1-u)} = C$. Thus, provided $\eta \neq \frac{1}{2}$, there is a unique $z^* = F^{-1}(u^*)$ such that $\frac{\tilde{\phi}'_\tau(F(z^*))}{\tilde{\phi}'_\tau(1-F(z^*))} = C$. One may verify that this $z^* > 0 \iff \eta > \frac{1}{2}$; for example, see Figure 4, which visualises the risk minimiser for various values of $\tau$. Thus, the sign of the minimising score conveys whether or not the positive class-probability is dominant; thus, the loss is classification calibrated.

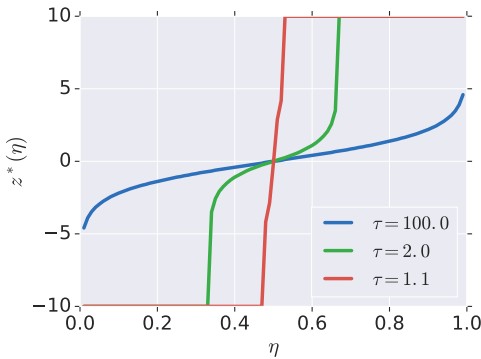

Figure 4: Risk minimiser for the partially Huberised logistic loss.

$\square$

*Proof of Proposition 7.* Let $R_\tau(f)$ denote the risk on the *clean* distribution of a predictor $f$ with respect to the partially Huberised loss $\tilde{\varphi}_\tau$ with parameter $\tau$. Similarly, let $\bar{R}_\tau(f)$ denote the risk on the *noisy* distribution. Following Ghosh et al. (2015); van Rooyen et al. (2015), we have

$$\bar{R}_\tau(f) = (1 - 2 \cdot \rho) \cdot R_\tau(f) + \alpha \cdot \underset{x}{\mathbb{E}} \left[ \tilde{\varphi}_\tau(f(x)) + \tilde{\varphi}_\tau(1 - f(x)) \right].$$

That is, the risk on the noisy distribution equals a scaled version of the risk on the clean distribution, plus an additional term. This term is a constant independent of $f$ if and only if $\tilde{\varphi}_\tau$ satisfies the symmetry condition of Ghosh et al. (2015), namely, $\tilde{\varphi}_\tau(u) + \tilde{\varphi}_\tau(1 - u) = C$ for some constant $C$.

Even when the symmetry condition does not hold, one may nonetheless aim to bound this additional term as follows. For simplicity, we restrict attention here to $\varphi$ being the log or cross-entropy loss $\varphi(u) = -\log u$. By definition, for any $f \in (0,1)$,

$$\tilde{\varphi}_\tau(f) + \tilde{\varphi}_\tau(1-f) = \begin{cases} -\tau \cdot f + \log \tau + 1 - \log(1-f) & \text{if } p \leq \frac{1}{\tau} \wedge \left(1 - \frac{1}{\tau}\right) \\ -\tau \cdot f - \tau \cdot (1-f) + 2 \cdot \log \tau + 2 & \text{if } 1 - \frac{1}{\tau} \leq p \leq \frac{1}{\tau} \\ -\log f - \log(1-f) & \text{if } \frac{1}{\tau} \leq p \leq \left(1 - \frac{1}{\tau}\right) \\ -\log f - \tau \cdot (1-f) + \log \tau + 1 & \text{if } p \geq \frac{1}{\tau} \vee \left(1 - \frac{1}{\tau}\right) \end{cases}$$

$$= \begin{cases} -\tau \cdot f + \log \tau + 1 - \log(1-f) & \text{if } p \leq \frac{1}{\tau} \wedge \left(1 - \frac{1}{\tau}\right) \\ -\tau + 2 \cdot \log \tau + 2 & \text{if } 1 - \frac{1}{\tau} \leq p \leq \frac{1}{\tau} \\ -\log f - \log(1-f) & \text{if } \frac{1}{\tau} \leq p \leq \left(1 - \frac{1}{\tau}\right) \\ -\log f - \tau \cdot (1-f) + \log \tau + 1 & \text{if } p \geq \frac{1}{\tau} \vee \left(1 - \frac{1}{\tau}\right). \end{cases}$$

Evidently, all piecewise functions involved are bounded on the respective intervals. For example,

$$\tilde{\varphi}_\tau(f) + \tilde{\varphi}_\tau(1-f) \leq \begin{cases} \log \tau + 1 - \log \left[\frac{1}{\tau} \vee \left(1 - \frac{1}{\tau}\right)\right] & \text{if } p \leq \frac{1}{\tau} \wedge \left(1 - \frac{1}{\tau}\right) \\ -\tau + 2 \cdot \log \tau + 2 & \text{if } 1 - \frac{1}{\tau} \leq p \leq \frac{1}{\tau} \\ -2 \cdot \log \frac{1}{\tau} & \text{if } \frac{1}{\tau} \leq p \leq \left(1 - \frac{1}{\tau}\right) \\ -\log \left[\frac{1}{\tau} \vee \left(1 - \frac{1}{\tau}\right)\right] + \log \tau + 1 & \text{if } p \geq \frac{1}{\tau} \vee \left(1 - \frac{1}{\tau}\right). \end{cases}$$

By taking the maximum of each of the quantities on the right hand side – which are constants depending on $\tau$ – we may thus find constants $C_1, C_2$ such that

$$\bar{R}_\tau(f) \leq (1 - 2 \cdot \rho) \cdot R_\tau(f) + C_1$$

$$\bar{R}_\tau(f) \geq (1 - 2 \cdot \rho) \cdot R_\tau(f) + C_2.$$

Now let $f^*$ denote the minimiser of the clean risk $R_\tau(f)$, and $\bar{f}^*$ the minimiser of the noisy risk $\bar{R}_\tau(f)$. Then, using each of the above inequalities,

$$\bar{R}_\tau(f^*) - \bar{R}_\tau(\bar{f}^*) \leq (1 - 2 \cdot \rho) \cdot (R_\tau(f^*) - R_\tau(\bar{f}^*)) + C_1 - C_2 \leq (C_1 - C_2),$$

where the last inequality is because $R_\tau(f^*) \leq R_\tau(\bar{f}^*)$ by definition of $f^*$. The claim follows. $\square$

# B    PROPER AND PROPER COMPOSITE LOSSES

Beyond requiring classification-calibration, it is often desirable to use classifier outputs as valid probabilities. *Proper losses* (Savage, 1971; Schervish, 1989; Buja et al., 2005) $\varphi\colon \{\pm 1\}\times [0,1] \to \bar{\mathbb{R}}_+$ are the core losses of such class-probability estimation tasks, for which

$$(\forall p \in [0,1])\,\underset{u\in[0,1]}{\operatorname{argmin}}\,\underset{y\sim\mathrm{Bern}(p)}{\mathbb{E}}\,[\varphi(y,u)] = p. \tag{13}$$

Equation 13 stipulates that when using $\varphi$ to distinguish positive and negative labels, it is optimal to predict the positive class-probability.

Typically, it is more useful to work with losses that accept real-valued scores, e.g. as output by the pre-activation of the final layer of a neural network. To this end, *proper composite* losses (Reid & Williamson, 2010; Nock & Nielsen, 2009) are those $\ell\colon \{\pm 1\}\times\mathbb{R} \to \mathbb{R}_+$ are such that $\ell\circ F^{-1}$ is proper, for invertible *link function* $F\colon \bar{\mathbb{R}} \to [0,1]$. Proper composite losses are inherently classification-calibrated (Reid & Williamson, 2010, Theorem 16). Canonical examples are the logistic loss $\ell(y,v) = \log(1 + e^{-yv})$ with $F\colon v \mapsto \sigma(v)$, and exponential loss $\ell(y,v) = e^{-yv}$ with $F\colon v \mapsto \sigma(2\cdot v)$. When $\ell$ is differentiable, we have (Reid & Williamson, 2010, Corollary 12)

$$(\forall v \in \mathbb{R})\, F(v) = \left(1 - \frac{\ell'(+1,v)}{\ell'(-1,v)}\right)^{-1}. \tag{14}$$

Given a proper loss $\varphi$ and "symmetric" link $F$ with $F(-v) = 1 - F(v)$, the loss $\ell(y,v) = \varphi(y, F(v))$ defines a margin loss (Reid & Williamson, 2010, Corollary 14).

Proper composite losses may also be extended to multiclass settings in the natural way (Gneiting & Raftery, 2007; Williamson et al., 2016): one now defines a proper loss $\varphi\colon [K] \times \Delta_K \to \mathbb{R}_+$, where $K$ is the number of classes and $\Delta_K$ denotes the $K$-simplex. A proper composite loss may be defined using a link $F\colon \mathbb{R}^K \to \Delta_K$, such as the softmax operator $F_k(\mathbf{v}) = \frac{\exp(\mathbf{v}_k)}{\sum_{k'}\exp(\mathbf{v}_{k'})}$. Combined with the log-loss $\varphi(y,p) = -\log p_y$, this yields the standard softmax cross-entropy loss.

## C ILLUSTRATION OF LOSSES

We illustrate the Huberised, partially Huberised, and generalised cross-entropy losses as their underlying tuning parameters are varied. Additionally, we illustrate the link functions that are implicit in each of the losses, which illustrates that they may be non-invertible if $\tau$ is too large.

### C.1 HUBERISED LOSS

Figure 5 illustrates the Huberised version of the logistic loss, and its derivative. Following the proof of Lemma 3, for $\tau \in (0, 1)$ and $z_0 = -\sigma^{-1}(\tau)$, the Huberised loss $\bar{\phi}_\tau$ has an implicit link function (see Figure 6)

$$(\forall z \in \mathbb{R}) \, \bar{F}_\tau(z) = \begin{cases} \frac{\sigma(z)}{\sigma(z_0)+\sigma(z)} & \text{if } z \leq z_0 \\ \sigma(z) & \text{if } z_0 \leq z \leq -z_0 \\ \frac{1}{2} & \text{if } -z_0 \leq z \leq z_0 \\ \frac{\sigma(z_0)}{\sigma(z_0)+\sigma(z)} & \text{if } z \geq -z_0. \end{cases}$$

Compared to the standard sigmoid, the Huberised link saturates more slowly as $\tau$ is decreased. Note that when $\tau \leq \frac{1}{2}$, the link function is not invertible everywhere: this results in the loss not being proper composite per our definition.

### C.2 PARTIALLY HUBERISED LOSS

Figure 7 illustrates the partially Huberised version of the logistic loss, as well as the base log-loss. Following the proof of Lemma 6, the partially Huberised loss $\tilde{\phi}_\tau$ has implicit link function

$$\tilde{F}_\tau(v) = \begin{cases} \frac{1}{1+\tau \cdot \sigma(-v)} & \text{if } \sigma(v) \leq \frac{1}{\tau} \wedge \left(1 - \frac{1}{\tau}\right) \\ \sigma(v) & \text{if } \frac{1}{\tau} \leq \sigma(v) \leq \left(1 - \frac{1}{\tau}\right) \\ \frac{1}{2} & \text{if } \left(1 - \frac{1}{\tau}\right) \leq \sigma(v) \leq \frac{1}{\tau} \\ \frac{\tau \cdot \sigma(v)}{1+\tau \cdot \sigma(v)} & \text{if } \sigma(v) \geq \left(1 - \frac{1}{\tau}\right) \vee \frac{1}{\tau}. \end{cases}$$

Figure 9 illustrates the link function. The logistic loss has $\varphi(u) = -\log u$, and so $\varphi'(1/2) = -2$. For $\tau = 1$, the link function will be non-invertible everywhere, which is expected since the loss here is the linear loss, which is not suitable for class-probability estimation. For $\tau \in (1, 2)$, the link function will be invertible for $p \notin \left[\left(1 - \frac{1}{\tau}\right), \frac{1}{\tau}\right]$. Intuitively, the case $\tau \in (1, 2)$ corresponds to the linear regions of the losses on positive and negative instances crossing over. For $\tau \geq 2$, the link function will always be invertible.

It may be observed that partial Huberisation causes the link function to saturate at values $[1/(1 + \tau), \tau/(1+\tau)]$: this does not affect classification calibration, but does imply that rescaling is necessary in order to intepret the output probabilities.

### C.3 ILLUSTRATION OF GENERALISED CROSS-ENTROPY

Figure 8 illustrates the base $\varphi_\alpha$ loss, and its composition with a sigmoid link function.

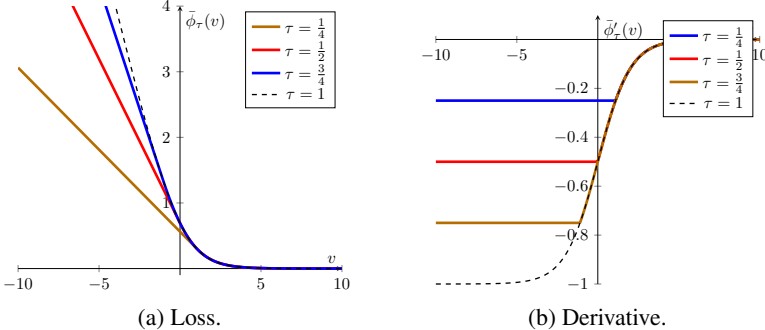

(a) Loss.

(b) Derivative.

Figure 5: Illustration of Huberised logistic loss. The effect of Huberisation is to linearise the loss beyond a certain point, or equally, to cap the derivative when its magnitude exceeds a certain threshold.

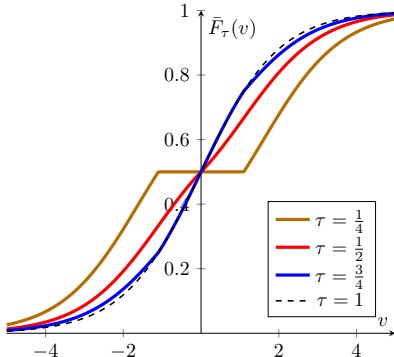

Figure 6: Illustration of link function for the Huberised logistic loss. The effect of Huberisation is to make the link function saturate slower. Note that when $\tau \leq \frac{1}{2}$, the link function is not invertible everywhere.

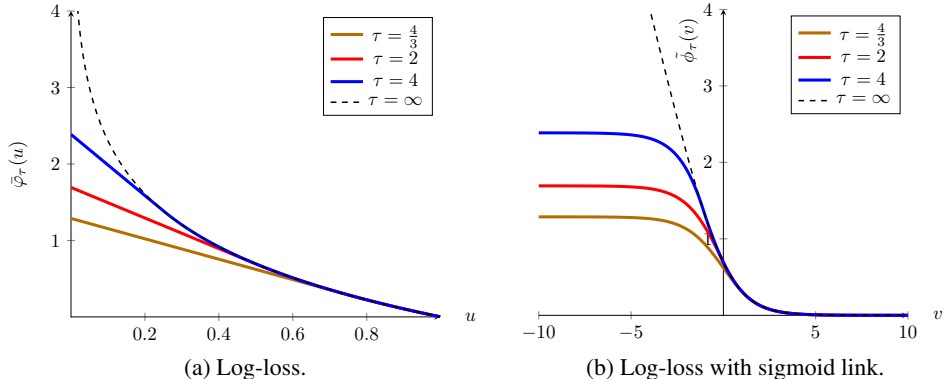

(a) Log-loss.

(b) Log-loss with sigmoid link.

Figure 7: Illustration of partial Huberisation, in terms of the underlying proper loss and its composite form.

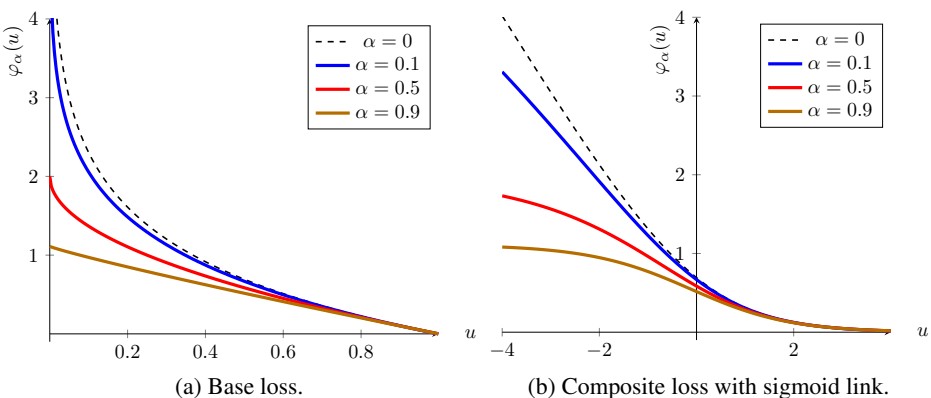

Figure 8: Illustration of generalised cross-entropy loss for various choices of $\alpha \in [0, 1)$.

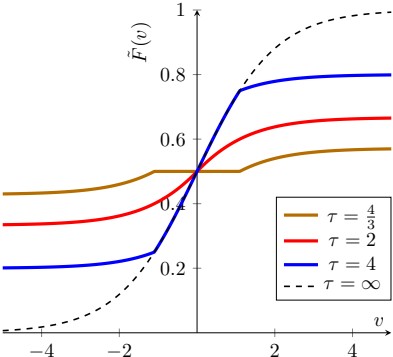

Figure 9: Illustration of link function for partially Huberised logistic loss. The effect of partial Huberisation is to make the link function saturate slower. When $\tau < 2$, the link function is not invertible everywhere.

# D    ROBUSTNESS TO INSTANCE OUTLIERS

Following Ding (2013, Section 3.2.2), we consider robustness to an arbitrary corruption under a linear scoring model. For margin loss $\ell(x, y; \theta) \doteq \phi(y \cdot \theta^{\mathrm{T}} x)$ with empirical risk minimiser $\hat{\theta}_N$ on a sample $\{(x_n, y_n)\}_{n=1}^N$, suppose that the sample is corrupted with an outlier $(x', y')$. One would like to ensure that $\hat{\theta}_N$ is not swayed by making $\|x'\|_2$ arbitrarily large: that is,

$$\lim_{\|x'\|_2 \to +\infty} \frac{1}{N+1} \sum_{n=1}^N \nabla \ell(x_n, y_n; \hat{\theta}_N) + \nabla \ell(x', y'; \hat{\theta}_N) = 0,$$

or equivalently, $\lim_{\|x'\|_2 \to +\infty} \nabla \ell(x', y'; \hat{\theta}_N) = 0$. Fortunately, the saturating behaviour of the partially Huberised loss affords this, as we now show.

**Proposition 8.** *Pick any convex, differentiable, proper composite margin loss $\phi$, whose link $F$ satisfies $\lim_{z \to -\infty} z \cdot F'(z) = 0$. For any $\tau > 0$, let $\tilde{\ell}_\tau(x, y; \theta) \doteq \tilde{\phi}_\tau(y \cdot \theta^{\mathrm{T}} x)$ be the $\tau$-partially Huberised loss under a linear model. Then, for any $(x, y)$ and $\theta$ such that $\|\theta\|_2 < +\infty$ and $\theta^{\mathrm{T}} x \neq 0$,*

$$\lim_{\|x\|_2 \to +\infty} \|\nabla \tilde{\ell}_\tau(x, y; \theta)\|_2 = 0.$$

*Proof of Proposition 8.* By definition,

$$\nabla \tilde{\ell}_\tau(x, y; \theta) = y \cdot x \cdot \phi'(y \cdot \theta^{\mathrm{T}} x).$$

Thus,

$$
\begin{aligned}
\|\nabla \tilde{\ell}_\tau(x, y; \theta)\|_2 &= \|x\|_2 \cdot |\phi'(y \cdot \theta^{\mathrm{T}} x)| \\
&= (y \cdot \theta^{\mathrm{T}} x) \cdot |\phi'(y \cdot \theta^{\mathrm{T}} x)| \cdot \frac{1}{y \cdot \|\theta\|_2 \cdot \cos \psi_{x,\theta}},
\end{aligned}
$$

where $\psi_{x,\theta}$ denotes the angle between $x$ and $\theta$. If $\theta^{\mathrm{T}} x \neq 0$, then $\cos \psi_{x,\theta} \neq 0$, and so the third term is finite. Thus, $\|\nabla \tilde{\ell}_\tau(x, y; \theta)\|_2 \to \lim_{z \to \pm\infty} |z| \cdot \tilde{\phi}'_\tau(z)$, depending on the sign of $y \cdot \theta^{\mathrm{T}} x$. By definition of $\tilde{\phi}_\tau$, the derivative of the loss asymptotes to either $F'(v)$, or $\phi'(v)$. Now, $\lim_{z \to -\infty} z \cdot F'(v) = 0$ by assumption, and $\lim_{z \to +\infty} z \cdot \phi'(v) = 0$ since $\phi$ is convex, the claim is shown.  $\square$

# E  ADDITIONAL SYNTHETIC EXPERIMENTS

We provide some more details regarding the synthetic data used in the body.

## E.1  DETAILS OF LONG & SERVEDIO (2010) DATASET

The problem considered in Long & Servedio (2010) (see Figure 10) comprises a distribution concentrated on six atoms $\{\pm(1, 0), \pm(\gamma, 5\gamma), \pm(\gamma, -\gamma)\} \subset \mathbb{R}^2$ for some $\gamma > 0$; we chose $\gamma = \frac{1}{24}$. An instance $(x_1, x_2)$ is labelled as $y = [\![x_1 \geq 0]\!]$. The instances are weighted so that the first four atoms have probability mass $\frac{1}{8}$, and the last two atoms mass $\frac{1}{4}$. We modify this distribution slightly by treating the atoms as means of isotropic Gaussians, and treating the marginal distribution over instances to be a mixture of these Gaussians with mixing weights given by the corresponding probability masses of the atoms.

## E.2  DETAILS OF OUTLIER DATASET

For the experiment involving outliers in feature space, the data comprises points on the line. Positively labelled samples are drawn from a unit variance Gaussian centered at $(1, 1)$, with positively labeled outliers drawn from $(-200, 1)$. Negatively labelled samples comprise the negation of all points. We learn an unregularised linear classifier from this data, which corresponds to a single scalar $\theta$.

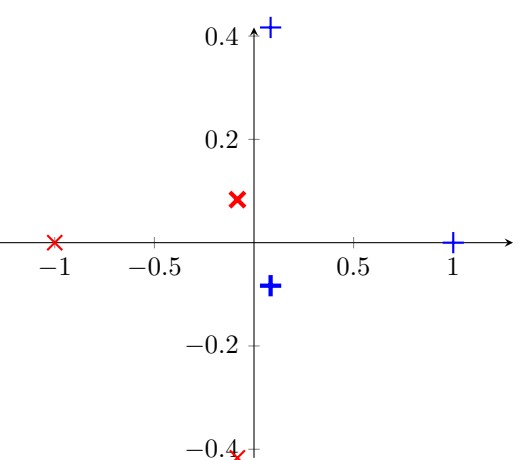

Figure 10: Distribution proposed in Long & Servedio (2010), which defeats any member of a broad family of convex losses. The data comprises six points, with the blue points labelled positive, and the red points labelled negative. The two "fat" points have twice as much probability mass as their "thin" counterparts. While the dataset is trivially linearly separable, minimising a broad range of convex losses with a linear model under any non-zero amount of symmetric label noise results in a predictor that is tantamount to random guessing.

