# OpenReview forum: "Can gradient clipping mitigate label noise?"
_ICLR.cc/2020/Conference — Accept (Poster)_

### Official Review · AnonReviewer2 · 2019-10-21
**Official Blind Review #2**

**Rating:** 8

**Review:**

This paper studied a fundamental problem in robust machine learning, that is, can gradient clipping mitigate label noise? The paper is well written, clearly motivated, highly novel and significant not only in a theoretical sense but also in a practical sense.

As argued in the abstract, gradient clipping is a widely-used technique which is generally motivated from the OPTIMIZATION point of view. In this paper, the authors proposed an entirely new motivation of gradient clipping from the ROBUSTNESS point of view, since intuitively it should be able to mitigate label noise. Surprisingly, the authors proved that for some simple binary classification with label noise, standard gradient clipping does not provide robustness; on the other hand, a simple variant of gradient clipping is robust, and is equivalent to suitably modifying the underlying loss function.

The major contributions were well summarized in the bottom of page 1, the related work was discussed in sec 4.3, and the caveats of this new methodology were also given in sec 4.4. Note that the proposed composite loss-based gradient clipping is applicable even on top of existing noise-robust losses, for example, the generalized cross-entropy loss, and this serves as a convincing demonstration of the great significance of the paper. Actually, the paper is full of insights, and I really enjoyed reviewing it.

I have a few questions on Tables 1 and 2. Table 1 said the standard gradient clipping is not robust according to Proposition 4. However, Proposition 4 is for the loss-based gradient clipping in (7) rather than the standard gradient clipping in (6). Perhaps I have missed something, but why the non-robustness of (7) can imply the non-robustness of (6)?

In Table 2, Linear loss on MNIST, the test accuracy was 9.6 when rho=0.6. Is this a typo? The accuracy was still 78.8 when rho=0.4. Moreover, the authors explained why the theoretically grounded linear loss performed so badly, that is, the optimization was so difficult. How about CE+clipping loss on CIFAR-100? The standard gradient clipping was also harmful here, even when there was no label noise at all. Was this due to bad optimization or robustness (or both)?

Since distributionally robust supervised learning is a future direction to go, I think the authors may be interested in a thought-provoking paper:
W. Hu, G. Niu, I. Sato, and M. Sugiyama. Does distributionally robust supervised learning give robust classifiers? ICML 2018.

**Experience Assessment:**

I have published in this field for several years.

**Review Assessment: Checking Correctness Of Derivations And Theory:**

I assessed the sensibility of the derivations and theory.

**Review Assessment: Checking Correctness Of Experiments:**

I carefully checked the experiments.

**Review Assessment: Thoroughness In Paper Reading:**

I read the paper thoroughly.

---

> ### Author Response · Authors · 2019-11-09
> **Response to AnonReviewer2**
>
> Thanks for the feedback!
>
> Regarding Proposition 4: you are correct that it concerns loss-based gradient clipping, which is not the same as gradient clipping in general. Following the comments to AnonReviewer3, the two are related for linear models when the norms of the inputs are constant; see the discussion following Equation 7.
>
> The results of linear loss on MNIST under high noise are not a typo — we observed that the linear loss can sometimes be unstable under high noise levels. In particular, the training can completely collapse. We hypothesise that one might be able to cope with this by adjusting learning rate schedule, for example, but in the interest of consistency we use the same architecture and settings regardless of noise level.
>
> Regarding the performance of CE + clipping on CIFAR-100, our intuition is the following: per Lemma 1, clipping is similar to linearising the base loss beyond a certain point. (It is not exactly so owing to the gradient of the margin.) Now, we observed that the linear loss does not perform well on CIFAR-100. One possible explanation for this is given in Sec 4.3, i.e., the gradients with respect to the probability estimates do not contain information about sample importances. We thus believe this poor performance translates to clipping with low enough thresholds.
>
> The citation on distributionally-robust learning is appreciated -- this is indeed relevant, and we have added it to the conclusion.

---

### Official Review · AnonReviewer1 · 2019-10-22
**Official Blind Review #1**

**Rating:** 6

**Review:**

Summary:

Gradient clipping has been studied as an optimization technique and also as a tool for privacy preserving, but in this paper, it studies the robustness properties of gradient clipping.  More specifically, the main question of the paper is: Can gradient clipping mitigate label noise?  The paper reveals that the answer is no, but further proposes a simple variant of gradient clipping is robust and has nice property of classification calibration.  Experiments show that the proposed variant works under label noise.


Strength of the paper:

The motivation and goal of the paper is stated in the title and is very clear, making it easier to follow the story of the paper.  There are sufficient background on the loss functions and gradient clipping in the beginning that helps guide the reader.  The proposed method is robust to label noise and has theoretical guarantees.  The relationship between similar work is summarized.  Experiments have both synthetic and benchmark datasets to demonstrate the behavior of the proposed method.


Weakness of the paper:

Currently, the experiments only include methods studied in the paper.  It would be better to include baseline methods stated in the end of Section 5 or in Section 4.3.

After response:
Thank you for the clarification!  I have read the other reviews and author comments.

**Experience Assessment:**

I have read many papers in this area.

**Review Assessment: Checking Correctness Of Derivations And Theory:**

I assessed the sensibility of the derivations and theory.

**Review Assessment: Checking Correctness Of Experiments:**

I assessed the sensibility of the experiments.

**Review Assessment: Thoroughness In Paper Reading:**

I read the paper at least twice and used my best judgement in assessing the paper.

---

> ### Author Response · Authors · 2019-11-09
> **Response to AnonReviewer1**
>
> Thanks for the feedback!
>
> We compare against the generalised cross-entropy (GCE) as this is (to our knowledge) a state-of-the-art loss-based technique for coping with label noise. Note that the methods discussed in Sec 4.3 employ the same base loss as GCE.
>
> The methods listed at the end of Sec 5 are based on distinct, complementary ideas. While certainly combining these with our clipped loss would be of interest, we wished to focus on our main message in the experiments (namely, the study of the viability of clipping to mitigate label noise).

---

### Official Review · AnonReviewer3 · 2019-10-23
**Official Blind Review #3**

**Rating:** 6

**Review:**


[Summary]
This paper studies the relationship between gradient clipping in stochastic gradient descent and robustness to label noise. Theoretical results show that gradient clipping in general is not robust to symmetric label noise. The paper then proposes a variant of gradient clipping (cl-clipping) that induces label noise robustness. Experiments support these claims on synthetic datasets and typical classification benchmarks.

[Decision]
The first contribution, that gradient clipping does not induce robustness to label noise, is an important negative result given the prominence of gradient clipping and datasets with noisy labels. The second contribution, cl-clipping, amounts to minimizing a non-convex loss with saturating regions but, as far as I know, these properties are necessary for robustness to label noise. Theoretical results are limited to SGD with mini-batch size 1 but the insights carry over to larger mini-batches in the experiments. Overall, I recommend acceptance.

[Comments]
The parameter tau controls robustness, and a higher noise level requires a higher tau. There is little discussion on how this parameter is chosen in the experiments. On the synthetic dataset, the Huberized loss uses tau=1 and the partially Huberized loss uses tau=2. How are these values chosen? Did the authors observe a U-shaped curve when sweeping over tau? On the real-world datasets, tau is fixed for each method across different noise levels. Does this mean that a single value of tau worked best regardless of the noise level, or was it tuned for a particular noise level?

Proposition 4 shows that symmetric noise breaks down the clipping method in Eq (7) which can be seen as a special case of gradient clipping. I might be missing something here, but it is not obvious to me that, when the norm of x is constant across the samples, Eq (7) is equal to gradient clipping.


**Experience Assessment:**

I have read many papers in this area.

**Review Assessment: Checking Correctness Of Derivations And Theory:**

I assessed the sensibility of the derivations and theory.

**Review Assessment: Checking Correctness Of Experiments:**

I carefully checked the experiments.

**Review Assessment: Thoroughness In Paper Reading:**

I read the paper at least twice and used my best judgement in assessing the paper.

---

> ### Author Response · Authors · 2019-11-09
> **Response to AnonReviewer3**
>
> Thanks for the feedback!
>
> We have updated the draft to include a discussion on the choice of τ in Section 5. This is a tuning parameter that can be set by the user, similar to the q parameter from generalised cross-entropy, or noise estimates in loss-correction techniques.
>
> In our experiments, we chose τ so as to maximise accuracy on a set of (noisy) validation samples for the setting of noise rate ϱ = 0.6. We did not tune τ for each noise rate, since this value of τ worked well for lower noise rates as well. Of course, there is no conceptual issue with tuning τ for each value of ϱ, although there is a slight computational cost.
>
> For the range of τ, we found that values of τ = { 2, 10 } — which correspond to clipping at probability values less than 0.5 and 0.1 respectively — generally gave good performance for the datasets considered. In general, one can certainly expand this range, again with a slight computational cost.
>
> We indeed observe a U-shaped curve in general. Note also that as τ gets closer to 1, we essentially reduce to the original loss (e.g., log-loss). As τ gets larger, we essentially reduce to the linear loss. As shown in Table 2, intermediate values of τ yield better performance than either extreme under label noise. We have made a comment on this following the discussion of how τ is chosen.
>
> Proposition 4 indeed concerns loss-based gradient clipping, i.e., Equation 7. Per the discussion following Lemma 1, this is not exactly the same as gradient clipping in general. However, note that for linear models, the term ∇mθ(x, y) = ||x||2, so if this is constant then Equations 6 and 7 will coincide. We have added some clarifying text after Equation 7.

---

### Decision · Program_Chairs · 2019-12-19

**Decision:**

Accept (Poster)

**Comment:**

This paper studies the effect of clipping on mitigating label noise. The authors demonstrate that standard gradient clipping does not suffice for achieving robustness to label noise. The authors suggest a noise-robust alternative. In the discussion the reviewers raised some interesting questions and technical detailed but mostly agreed that the paper is well-written with nice contributions. I concur with the reviewers that this is a nicely written paper with good contributions. I recommend acceptance but recommend the authors continue to improve their paper based on the reviewers' suggestions.